# ESG growth catalyst: China's Central Bank collateral framework expansion

Xupei Wang, Liang Zhao*

School of Economics, Guangdong Ocean University, Zhanjiang, Guangdong, China

* zhao.liang@gdou.edu.cn

## Abstract

Environmental, social, and governance (ESG) plays an important role in corporate sustainability. In June 2018, the People's Bank of China (China's central bank) decided to expand the scope of the Medium-Term Lending Facility (MLF) collateral to include green bonds issued by companies. This study investigates the impact of the PBOC's policy of integrating green bonds into the collateral framework on corporate ESG performance, specifically by alleviating financing constraints and fostering green innovation. Specifically, this study takes the Expanded Pledge Framework Policy (EPCF) of the Central Bank as a quasi-natural experiment and uses the panel data of China's A-share listed companies from 2014 to 2022 to investigate the causal relationship between EPCF and corporate ESG performance through the differential method (DID) and other methods. The results show that EPCF can improve the ESG performance of enterprises by easing the financing constraints of enterprises and promoting the green technology innovation of enterprises. On average, implementing an EPCF improves the relative ESG performance of an organization by about 0.02 units. In addition, through the parallel trend test and the results of existing literature, it can be seen that although enterprises issuing green bonds can improve ESG performance, the range is still limited, and EPCF has a greater improvement in the ESG performance of non-heavy polluting enterprises. More importantly, the implementation of EPCF is more effective in regions with a higher degree of marketization and in coastal areas. Our model has withstood extensive robustness checks.

## 1. Introduction

Global climate change and environmental pollution are becoming increasingly severe, prompting companies are turning their attention to the development of sustainable development, social responsibility, and labor rights. The international community, including governments, organizations, enterprises, and stakeholders—has been paying greater attention to the "environmental, social and Governance (ESG) system [1–3]. The reason for the heightened interest in ESG is that numerous

**Data availability statement:** All relevant data are within the manuscript and its Supporting Information files.

**Funding:** Guangdong Province Philosophy and Social Science Planning Project (GD24XWL01); Research project funded by the Guangdong Ocean University Scientific Research Startup Fund: "Study on the Transmission Mechanism of Imported Financial Crises" The funders had no role in study design, data collection and analysis, decision to publish, or preparation of the manuscript.

**Competing interests:** The authors have declared that no competing interests exist.

organizations and studies have demonstrated a strong correlation between ESG practices and firm performance. For example, a McKinsey report analyzed 180 publicly listed companies and found that those with excellent ESG performance achieved better financial outcomes, with a return to shareholders that was six percentage points higher than the average. Additionally, the analysis revealed that companies excelling in ESG performance scored higher on social indicators, such as community relations, employee welfare, and supply chain management. These social ESG practices are positively correlated with corporate financial performance [4]. An increasing number of studies have demonstrated that ESG indicators serve as comprehensive measures of corporate sustainability, effectively addressing the three dimensions of environment, society, and corporate governance [5–9]. Concurrently, the initial implementation of corporate ESG practices requires corresponding supportive policies to facilitate progressive improvement. Following the global financial crisis of 2008–09, both employment and short-term policy interest rates fell sharply. In an effort to restore employment and stimulate the economy, the Federal Reserve and the European Central Bank, representing developed countries, sequentially altered their original monetary policy frameworks, notably by implementing a variety of innovative monetary policy tools. China, as a representative of the developing world, has also been impacted by the ongoing crisis. In response to these challenges, the People's Bank of China has significantly reformed its monetary policy framework since 2013 by introducing several new lending instruments. For instance, in 2014, the PBOC implemented the Medium-Term Lending Facility (MLF) policy, which requires eligible commercial banks to provide bond assets—such as treasury bonds, central bank bills, and corporate bonds with high credit ratings (i.e., AAA ratings)—as collateral in exchange for funding. Subsequently, on June 1, 2018, the PBOC announced an expansion of the medium-term lending facility (MLF) collateral scope to include green assets, such as green bonds and credits that meet specific criteria, in an effort to direct more funds toward the green industry [10]. This initiative will enable enterprises to concentrate on developing green industries while simultaneously engaging in traditional sectors, thereby upholding their commitment to environmental sustainability and social responsibility. The impact of this forward-thinking central bank guarantee policy on sustainable business operations warrants further analysis.

Although previous literature has examined how EPCF affect financial markets at the macro level [11], but there remains a gap in research regarding its impact on the sustainable development of the economy. Financial policy is the lifeblood of economic development, and effective financial policy enhances market efficiency [12]. This is evident in the various ways financial policies can influence economic development, such as alleviating financial stress [13], promoting employment growth [14] and fostering economic development through financial agglomeration [15]. In recent years, there has been a growing interest in corporate sustainability [16] As a barometer of the micro-economy, corporate sustainability underpins long-term macroeconomic development [17,18]. Therefore, it is essential to explore the impact of EPCF on corporate sustainability. On one hand, conducting the study at the micro level can alleviate the complexities associated with macro-level environmental issues, thereby

yielding a more accurate understanding of causal relationships. On the other hand, this approach can also uncover the mechanisms through which financial policies exert their effects—mechanisms that may be challenging to identify at the macro level. Overall, concentrating the study on corporate sustainable development can enhance existing theories related to financial policies and offer new insights for the future formulation of these policies. This also leads us to the first key question of this study:

Question 1: Does the EPCF impact firms' ESG performance?

In addition, the existing literature provides several insights into how financial policy influences enterprise development. First, financial policy can directly impact firms' financing costs and investment behavior by adjusting credit supply and interest rate levels [19]. And a loose credit environment can alleviate financing constraints for firms, thereby encouraging technological innovation [20].

Second, financial policies can significantly influence the degree of financialization within firms, which refers to the tendency of firms (i.e., firms allocate more resources to financial assets than to physical production) and thus their production. While moderate financialization can enhance resource allocation efficiency, excessive financialization may hinder enterprise development and technological innovation [21]. Furthermore, financial policies can directly facilitate the transition of enterprises toward sustainable development by supporting green projects [22]. Overall, there is a diverse array of ways in which financial policies impact firms. This leads us to the second key question of this study:

Question 2: If the EPCF has an impact on firms' ESG performance, what is the mechanism behind this impact?

In light of this, this paper uses data from A-share listed companies between 2014 and 2022, focusing on a natural experiment that occurred when green bonds were incorporated into the collateral framework in June 2018. The study employs double-differencing and other methodologies to investigate the effects of the expansion policy on corporate ESG performance and its underlying mechanisms. Specifically, the implementation of the EPCF is found to enhance corporate ESG performance by approximately 0.02 units. Furthermore, the EPCF can improve firms' ESG performance by alleviating their financing constraints, which in turn promotes green technology innovation and ultimately enhances their ESG performance. Additionally, this study reveals that the extent of EPCF's impact on firms' ESG performance is influenced by the firms' environmental characteristics, marketization levels, and geographic locations.

The potential marginal contributions of this study are numerous. First, unlike previous macro-level studies, we investigate the impact of EPCF on corporate sustainability at the micro level, thereby addressing the gap in the existing research on financial policies at this scale. Second, drawing on information asymmetry theory and scarcity theory, this study elucidates how EPCF influences firms' ESG performance. Consequently, it broadens the application scenario of these relevant theories. Third, the heterogeneity analysis, which considers both internal and external factors affecting enterprises, offers valuable insights for enhancing the current policy framework.

The structure of the remainder of this study is organized as follows: Section 2 reviews the existing literature and presents the research hypotheses derived from the theoretical framework. Section III outlines the data, sample, and methodology employed in this study. Section IV analyzes and interprets the primary empirical findings. Finally, Section V summarizes the conclusions and offers pertinent recommendations.

## 2. Literature review and theoretical hypotheses

### 2.1. Central bank collateral policy and corporate ESG performance

Collateral is one of the innovative products of modern finance, and its management mechanism is a crucial means of preventing and controlling financial risks. The central bank's collateral framework serves two primary roles: preventing systemic risks and facilitating counter-cyclical adjustments in practice [23]. Relevant studies illustrate that unconventional monetary policy, based on the collateral framework, has significant effects [24–26]. It mainly affects the collateral market

through the scarcity channel, the structural channel, or a combination of both [27], thereby reducing the cost of financing for firms seeking to expand their corporate financing scale [28]. As financing conditions ease, firms are likely to invest more and increase production, leading to employment growth and, concurrently, higher labor costs. With financial support, firms are able to recruit more skilled and educated employees, enhance productivity, and foster technological innovation, ultimately raising total factor productivity. Additionally, while MLF operations enhance liquidity, they also increase the willingness and capacity of firms to provide trade credit.

The determinants of corporate ESG performance have been extensively studied. From an internal governance perspective, factors such as alleviating financing constraints [29] digital transformation [30], employee insurance [31] higher education [32], informal board hierarchy structure [33] and corporate culture [34] all contribute to enhancing corporate ESG performance. Additionally, external pressures also contribute to improving corporate ESG performance, including environmental protection taxes [35], green finance policies [36], tax incentives [37] minimum wage standards [38], government subsidies [39], fiscal pressure [40] and bank competition [41].

## 2.2. The mechanism role of financing constraint and enterprise green innovation

Information Asymmetry Theory [42] suggests that the two parties involved in a transaction experience disparities in information due to factors such as the sequential nature of information acquisition, inconsistent access to information channels, and varying interpretations of information. In a real market economy, information is often neither open nor transparent. Enterprise managers may conceal negative information about their organizations, while stakeholders frequently cannot afford the high costs associated with obtaining complete information. As a result, stakeholders possess incomplete information. This information asymmetry raises concerns for investors regarding the financial status, corporate governance, and research and development (R&D) activities of firms, which they cannot fully grasp [43,44]. Especially in the face of R&D projects in the direction of sustainable development, such as green innovation, which have high investment risks and long payback cycles, commercial banks may be "reluctant to lend". This will not be conducive to innovation. Within the principal-agent relationship between enterprises and investors, information asymmetry presents a significant challenge. Investors often struggle to fully understand the operational status of enterprises, project prospects, and the actual utilization of funds. By expanding the collateral framework to include green bonds, central banks effectively send signals to the market, conveying information about the quality and risk associated with the collateral [45]. These signals indicate that green bonds are supported by the central bank's credit endorsement, which can significantly reduce the level of information asymmetry between firms and investors. Moreover, to qualify for central bank guarantee support, companies must also strive to enhance their ESG performance. Therefore, the central bank's expansion of the collateral framework to encompass green bonds will incentivize firms to adopt more environmentally friendly and responsible business strategies to align with investor expectations and market demand. Based on this analysis, Hypothesis 1 is proposed.

H1 Collateral framework expansion policies have a significant positive impact on corporate ESG performance

Based on the theory of scarcity [45], the inclusion of green bonds in the collateral framework by central banks allows financial institutions, such as commercial banks, to exchange these bonds for liquidity. This process renders green bonds a relatively scarce resource [45]. Consequently, the scarcity can stimulate a strong preference for green bonds among financial institutions, thereby enhancing the demand for their acquisition and alleviating the financing constraints faced by green firms. This premise leads to the formulation of green firms [46–48]. Based on this, hypothesis 2 is proposed.

H2a Collateral framework expansion policy enhances firms' ESG performance by alleviating their financing constraints.

Firms frequently encounter financing constraints when engaging in green innovation activities, primarily due to the uncertainty of returns, information asymmetry inherent in the innovation process, and elevated compliance costs. Consequently, these financing constraints can adversely affect firms' innovation activities [49]. And EPCF can provide firms

with the necessary guarantees to issue green bonds. As the volume of debt issuance continues to grow, the financing constraints faced by enterprises will be mitigated. This alleviation enables firms to secure adequate funding for the acquisition of environmental protection equipment and to enhance their investment in green R&D. Ultimately, this will lead to a significant improvement in the level of green innovation within these enterprises [50,51].

[52–55]. In addition, it has been found that corporate ESG performance can promote its own level of green innovation [56]. In fact, elevated levels of green innovation can enhance a company's ESG performance. There is a bidirectional causal relationship between the enhancement of green innovation and the improvement of firms' ESG performance. Specifically, green innovation facilitates the reduction of carbon emissions and the optimization of resource efficiency, thus enhancing their environmental governance capability [57]. Meanwhile, a high level of green innovation capability can enhance the social recognition of enterprises thus obtaining support from policies and external resources [58,59]. In summary, hypothesis 3 is proposed

H2b The collateral framework expansion policy enhances firms' ESG performance by promoting firms' green technology innovation.

## 2.3. Differences in central bank guarantee policies affecting ESG performance across samples

It has demonstrated that central bank monetary policy significantly influences firms' investment decisions by regulating the financing environment and associated costs [60]. Furthermore, increased liquidity resulting from monetary liquidity facilities (MLF) can alleviate credit constraints for non-state firms, with this effect being particularly pronounced in regions with lower levels of financial development [61]. In contrast, firms operating in more market-oriented regions benefit from enhanced investor protection, stricter legal enforcement, and less government intervention. This environment encourages firms to prioritize the reduction of agency costs and the improvement of investment efficiency, thereby fostering sustainable development and bolstering investor confidence [62]. In such contexts, firms are better positioned to perceive policy changes promptly and adjust their strategies accordingly. Therefore, the following hypothesis is proposed:

**H3a: The positive impact of policy implementation on firms' ESG performance is more pronounced in regions with a high level of marketization compared to regions with a low level of marketization**

The People's Bank of China has strengthened its support for green development by incorporating green assets into its collateral framework. Additionally, it has established a differentiated penalty mechanism for heavy polluters, which includes measures such as regulating credit scales and increasing financing costs [63]. This approach fosters an enabling environment for inter-firm mitigation efforts and stimulates innovation among non-heavy polluters [64], continuously improving the level and efficiency of green innovation, which in turn improves the performance of corporate ESG. Therefore, the following hypothesis is proposed:

H3b: The positive impact of implementing policies on corporate ESG performance is more pronounced for non-heavily polluting firms than for heavily polluting firms

Coastal regions possess natural port resources that facilitate international trade, thereby enhancing the connection between domestic and international markets and reducing logistics costs for enterprises. Consequently, these areas often experience industrial clustering. This clustering phenomenon suggests that firms are more likely to improve their ESG performance by learning from and emulating neighboring firms [65] Secondly, due to the developed economies of coastal regions attract significant attention from investors, media, and public attention. This heightened level of scrutiny creates a robust external monitoring mechanism. Effective external monitoring can compel enterprises to better serve their stakeholders, as increased external pressure encourages firms to enhance their performance and social responsibility, ultimately leading to improved ESG performance [66]. Therefore, based on the above analysis, the following hypothesis is proposed:

**H3c: Compared to the enterprises in the central and western regions, for the eastern enterprises the positive impact of the implementation of policies on the performance of corporate ESG is more pronounced**

## 3. Data and methodology

### 3.1. Methodological design

In order to effectively identify the impact of central bank EPCF on firms' ESG performance, this study treats the central bank EPCF as a quasi-natural experiment and empirically tests it using the DID methodology, which is capable of estimating the net effect of the policy through policy-induced changes in both the cross-sectional and time-series dimensions and which can alleviate the endogeneity problem to a certain extent when combined with regression analysis. The DID method has been widely used in policy research [67–71]. Specifically, this study sets up the DID model as shown in equation (1).

$$hzesg_{jt} = \alpha + \beta did_{jt} + control_{jt}\lambda' + ind_i + year_t + province_p + \varepsilon_{jt} \tag{1}$$

where i, j, t, and p denote industry, firm, year, and province identifiers, respectively. $hzesg_{jt}$ denotes the CSI ESG rating of the jth firm in year t; $\alpha$ denotes the intercept term; $\beta$ denotes the regression coefficients of the core explanatory variables to be focused on (i.e., the treatment effect of the policy); $did_{jt}$ denotes the dummy of the state of whether or not the jth firm is affected by the EPCF at year t variable (see Variable Selection and Construction below for specific construction); $\lambda'$ denotes the transpose of the row vector containing the regression coefficients of each control variable; $control_{jt}$ denotes the row vector containing the control variable at year t for the jth firm; $ind_i$ denotes the industry fixed effect; $year_t$ denotes the year fixed effect; $province_p$ denotes the province fixed effect; $\varepsilon_{jt}$ denotes the randomized disturbance term.

Although the DID method has many advantages, its use presupposes that there is no significant difference between the experimental and control groups before the intervention. Therefore, this study used the event study method to test for parallel trends between the experimental and control groups, as modeled in equation (2).

$$hzesg_{jt} = \alpha + \sum_{k=-4}^{T} \beta_k treat_j \times D_{kt} + ccontrol_{jt}\lambda' + ind_i + year_t + province_p + \varepsilon_{jt} \tag{2}$$

where i, j, t, and p denote industry, firm, year, and province identifiers, respectively. $\beta_k$ denotes the dynamic treatment effect in each period; $treat_j \times D_{kt}$ denotes the estimation of the treatment effect in the experimental and control groups in each period; the unputted pre-intervention interaction term $treat_j \times D_{kt}$ is used as a reference frame for the baseline to avoid strict multicollinearity. k means the gap between the current period and the time of the policy. The meaning of k is the gap between the current period time and the time when the policy was introduced. t is the set of all possible values (T = -4, -3, -2, 0, 1, 2, 3, 4). The rest of the symbols are the same as in equation (1).

To further examine the potential differences between samples with different characteristics, the heterogeneity analysis model (3), (4), and (5) were set up by interacting the heterogeneity variables with the core explanatory variables.

$$hzesg_{jt} = \alpha + \beta_1 market_{pt} \times did_{jt} + \beta_2 did_{jt} + \beta_3 market_{pt} + +control_{jt}\lambda' + ind_i + year_t + province_p + \varepsilon_{jt} \tag{3}$$

$$hzesg_{jt} = \alpha + \beta_1 polluted_{jt} \times did_{jt} + \beta_2 did_{jt} + \beta_3 polluted_{jt} + control_{jt}\lambda' + ind_i + year_t + province_p + \varepsilon_{jt} \tag{4}$$

$$hzesg_{jt} = \alpha + \beta_1 east_{jt} \times did_{jt} + \beta_2 did_{jt} + \beta_3 east_{jt} + control_{jt}\lambda' + ind_i + year_t + province_p + \varepsilon_{jt} \tag{5}$$

where i, j, t, and p denote industry, firm, year, and province identifiers, respectively. $hzesg_{jt}$ denotes the CSI ESG rating of the jth firm in year t; $\alpha$ denotes the intercept term; $\beta$ denotes the difference between the difference between the

experimental group and the control group to be focused on and the difference between the differences between heterogeneous subgroups; $market_{pt}$, $polluted_{jt}$, and $east_{jt}$ are the heterogeneity variables in (3), (4) and (5), respectively, and are all dummy variables. The specific construction method is described in the following explanation and construction of variables. The rest of the notation is the same as equation (1).

### 3.2. Variable selection and construction

**3.2.1. firm ESG performance.** The explanatory variable is corporate ESG performance. In this study, the CSI ESG composite score is used as a proxy variable to measure corporate ESG performance. As this dataset maximizes the sample and time span coverage of Chinese A-share listed companies, the data is time-sensitive. Therefore, the CSI ESG rating and score data have been widely used in many recent studies. To facilitate the interpretation of the estimated coefficients, they are divided by 100 as a measure of firms' ESG performance. The higher the ESG score, the better the firm's ESG performance [72–74].

**3.2.2. Central bank collateral framework expansion policy (EPCF).** The core explanatory variable is a dummy variable for whether firms are affected by the policy of expanding the central bank's collateral framework, and we use the term "policy" as a proxy in the following. Referring to the grouping idea of Wei et al.'s study [75]. In this paper, companies that have issued green bonds before 2018 and have maturity dates after 2018 (including 2018) are selected as the experimental group, and the rest of the listed companies are selected as the control group. In this study, Treat is a grouping dummy variable, and Treat takes 1 if the enterprise has issued green bonds before 2018 and the maturity date is after 2018 (with 2018 included), otherwise, it takes 0 (i.e., the one that takes 1 is the experimental group and the one that takes 0 is the control group). In addition, samples with green bond issuance dates and maturity dates both before 2018 or issuance dates and maturity dates both after 2018 are excluded to prevent the effect of the policy from being underestimated. Post is a time dummy variable, with the expansion of the central bank's collateral framework in 2018 as the policy point in time, and Post is taken to be 0 before 2018, and 1 after 2018 (with 2018 included.) Finally, the grouped dummy variables are cross-multiplied with the time dummy variables to obtain the core explanatory variable did.

**3.2.3. Control variables.** Relevant studies have shown that the larger and older a firm is, the higher its external visibility [76], the more it tends to disclose its ESG information to build a desirable image, and the more likely it is to engage in ESG practices [77,78]; return on total assets (ROA) measures the return on capital and is considered to be closely related to corporate ESG performance [79]; leverage ratio(LEV) measures the return on capital and is considered to be closely related to corporate ESG performance [58]; leverage measures a firm's short-term solvency, and a firm's reliance on debt financing may affect the performance of ESG firms because of higher investor visibility [80]; and firm profitability (Growth), a factor that can be used as a resource for ESG activities [81]. The shareholding structure may also influence a company's ESG behavior [82];board governance indicators like two-ranking influence ESG outcomes through advisory or supervisory role [83,84]; and, more importantly, state-owned enterprises (SOE) are under significant political pressure to take on environmental and social responsibility [85], which may lead to higher ESG performance of SOEs [86]. In addition, in drawing on Baldini, M. et al.'s study [80], this study introduces the provincial level of economic development (LNGDP) as a control variable for the region. To summarize, this study selected the following control variables: enterprise size (SIZE), enterprise years of experience (LNAGE), profitability (ROA), gearing ratio (LEV), two-job unity (DUA), whether state-controlled (SOE) equity concentration (DOMINANCE), and provincial economic development level (LNGDP). And the logarithmic treatment of enterprise years and provincial economic development level to obtain smoother data.

**3.2.4. Mechanism variables.**

1) Financing constraints

This study uses the SA index, which is more relevant than the traditional KZ index, to measure the financing constraints (SA) faced by firms. The SA index was proposed by Hadlock and Pierce in 2010 [87]. The SA index captures firms'

financing constraints from several perspectives, but it focuses on two key factors: firm size and firm age. In the SA index, larger and older firms are generally considered to face fewer financial constraints. This index provides a quantitative measure of the impact of firm size and time of establishment on its ability to raise finance. It has now been used in numerous studies [88–90].

2) Level of green innovation

Most studies use the total number of patents to measure the level of innovation [91], while the total number of green patents is commonly used to measure the the total number of green patents is often used to measure the level of green innovation, as the number of patents can be a more tangible output of the invention process [92]. It is worth mentioning that there are also studies that use indicators such as the number of green patent citations, green new product development, and green investment to measure the degree of green innovation of firms [93]. However, limited by factors such as data acquisition, after comprehensive consideration, this paper chooses the total number of green patents newly applied by enterprises each year to measure the level of green innovation of enterprises. In addition, in order to mitigate the effect of heteroskedasticity, the variable is logarithmic after adding 1.

### 3.2.5. Heterogeneity variables.

1) Marketization level

This study takes the index of marketization level measured by Fan Gang,Wang Xiaolu,Zhu and Heng,peng [93,94] The marketization index is supported by a system composed of multifaceted indicators. It reflects the progress of marketization in five aspects: the relationship between government and market, the development of a non-state economy, the degree of development of the product market, the degree of development of the factor market, the development of market intermediary organizations, and the legal and institutional environment. In this study, the marketization index is ranked from largest to smallest, with the top 20% of firms scored as 1 and the next 80% scored as 0. Therefore, it can fully take into account more Chinese localization factors.

2) Pollution intensity

Pollution intensity is a dummy variable in this study. According to the "Guidelines on Environmental Information Disclosure for Listed Companies" published by the Ministry of Environmental Protection in 2010 [95], enterprises belonging to heavily polluted industries are labeled as 1, and vice versa as 0.

3) Geographic location

Geographic location is a dummy variable in this study. In reference to the study by He et al [96], this study marks the seaward areas as 1. Conversely, non-seaward areas were recorded as 0.

### 3.3. Sample selection and data sources

This study utilizes a sample of A-share listed enterprises in mainland China from 2014–2022 as the research sample. Based on the following considerations: first, the data disclosure practices of A-share companies are standardized and encompass major industries, thereby reflecting the overall impact of the policy on the real economy; second, the period from 2014 to 2022 represents a critical phase in the evolution of the policy aimed at expanding the collateral capacity of green bonds from the germination to the deepening of the green bond, and the ESG rating system has progressively matured during this timeframe, providing substantial data support for analyzing the dynamic effects of the policy." The green bonds are sourced from the corporate green bond issuance records within the bond sector of the WIND database. The ESG performance data for the listed companies is derived from the CSI ESG composite score in the financial data in the stock sector of the WIND database. Enterprise-level data is obtained from the financial data segment of the

stock section of the WIND database, as well as from the enterprise financial data segment of the CSMAR database. Provincial-level data is extracted from the China Statistical Yearbook. Furthermore, the raw data underwent several processing steps: (1) exclude ST, PT, and *ST companies during the sample period; (2) delete samples with missing variables in the observations (3) delete listed companies in the financial industry. The paper finally obtained 27920 observations. The descriptive statistics of the processed data are presented in Table 1.

## 4. Empirical analysis

### 4.1. Benchmark regression

In order to avoid the impact of multicollinearity on the estimation results, the benchmark regression adopts the stepwise regression method. Specifically, on the basis of adding the core explanatory variables for regression, the control variables are gradually added and then regressed. If the significance of the core explanatory variables does not change significantly with the addition of the control variables, it can be assumed that the significance of the core explanatory variables is not affected by severe multicollinearity. Columns (1) through (10) in Table 1 report the stepwise regression process. In this study, the main focus is on the regression coefficients of the core explanatory variable *did*. As can be seen from the results in Table 2, the regression coefficients of the core explanatory variable *did* are still all significantly positive at the 1% level despite the fact that the significance of some of the explanatory variables may change during the sequential addition of the control variables. This indicates that the implementation of MIF (assuming that the policy is abbreviated as MIF) can effectively enhance the recognition of green bonds, which in turn improves the ESG performance of enterprises. And, on average, implementing MIF improves relative ESG performance by approximately 0.02 units after all control variables are included. In addition, ROA, SIZE and SOE are all significantly positive in their regression coefficients in the stepwise regression process, and LNAGE is all significantly negative in the stepwise regression process and it is statistically significant at the 1% level. There is no doubt that ROA, as one of the measures of a firm's profitability, influences the firm's ESG performance. High ROA firms focus more on long-term sustainable development, and ESG practice is one of the important dimensions. size, on the other hand, measures the size of a firm, which also affects the ESG performance. Larger firms tend to have more financial resources, meaning they have more money to invest in ESG practices. In addition, because companies with shorter history are more flexible and less bound by traditional business models, they are able to be more innovative and forward-looking in grasping market trends to a certain extent, and this acumen directly contributes to the fact that they are more likely to realize the importance of corporate ESG practices and thus take

**Table 1. Descriptive statistics.**

| VARIABLES | Obs | Mean | Std. Dev. | Min | Max |
|---|---|---|---|---|---|
| hzESG | 27920 | 0.732 | 0.053 | 0.572 | 0.845 |
| did | 27920 | 0.004 | 0.063 | 0 | 1 |
| LEV | 27920 | 0.413 | 0.201 | 0.058 | 0.886 |
| GROWTH | 27920 | 0.354 | 0.876 | -0.68 | 5.942 |
| DUA | 27920 | 0.313 | 0.464 | 0 | 1 |
| DOMINANCE | 27920 | 33.69 | 14.591 | 8.734 | 73.82 |
| ROA | 27920 | 0.037 | 0.064 | -0.257 | .2 |
| LNAGE | 27920 | 2.068 | 0.936 | 0 | 3.367 |
| SOE | 27920 | 0.321 | 0.467 | 0 | 1 |
| SIZE | 27920 | 22.264 | 1.292 | 19.99 | 26.326 |
| LNGDP | 27920 | 12.786 | 2.015 | 10.487 | 16.14 |
| SA | 27920 | -3.854 | 0.245 | -4.558 | -3.117 |
| GREENTECH | 27920 | 0.445 | 0.89 | 0 | 7.062 |

**Table 2. Benchmark regression.**

| VARIABLES | (1) | (2) | (3) | (4) | (5) | (6) | (7) | (8) | (9) | (10) |
|---|---|---|---|---|---|---|---|---|---|---|
| | hzESG | hzESG | hzESG | hzESG | hzESG | hzESG | hzESG | hzESG | hzESG | hzESG |
| did | 0.031*** | 0.035*** | 0.035*** | 0.035*** | 0.035*** | 0.033*** | 0.034*** | 0.034*** | 0.020*** | 0.020*** |
| | (4.59) | (5.46) | (5.55) | (5.47) | (5.61) | (4.85) | (4.98) | (5.31) | (3.03) | (3.03) |
| Lev | | 0.030*** | 0.030*** | 0.030*** | 0.030*** | 0.009* | -0.003 | -0.007 | 0.045*** | 0.045*** |
| | | (-5.87) | (-5.83) | (-6.03) | (-6.28) | (-1.86) | (-0.74) | (-1.65) | (-17.41) | (-17.43) |
| GROWTH | | | -0.002 | -0.002 | -0.002 | -0.002** | -0.002** | -0.002** | -0.001 | -0.001 |
| | | | (-1.63) | (-1.63) | (-1.64) | (-2.23) | (-2.33) | (-2.67) | (-1.40) | (-1.40) |
| DUA | | | | -0.003*** | -0.002*** | -0.002*** | -0.004*** | -0.001 | -0.000 | -0.000 |
| | | | | (-3.25) | (-3.32) | (-3.34) | (-5.07) | (-1.32) | (-0.48) | (-0.49) |
| DOMINANCE | | | | | 0.000*** | 0.000*** | 0.000*** | 0.000*** | 0.000 | 0.000 |
| | | | | | (8.04) | (5.75) | (4.96) | (3.33) | (0.63) | (0.63) |
| ROA | | | | | | 0.180*** | 0.175*** | 0.173*** | 0.114*** | 0.114*** |
| | | | | | | (18.82) | (19.17) | (20.52) | (17.37) | (17.36) |
| LNAGE | | | | | | | -0.004*** | -0.007*** | -0.012*** | -0.012*** |
| | | | | | | | (-6.49) | (-14.14) | (-16.56) | (-16.56) |
| SOE | | | | | | | | 0.019*** | 0.014*** | 0.014*** |
| | | | | | | | | (8.92) | (6.54) | (6.53) |
| SIZE | | | | | | | | | 0.014*** | 0.014*** |
| | | | | | | | | | (37.60) | (37.64) |
| LNGDP | | | | | | | | | | -0.000 |
| | | | | | | | | | | (-0.67) |
| Constant | 0.732*** | 0.744*** | 0.745*** | 0.746*** | 0.731*** | 0.720*** | 0.728*** | 0.733*** | 0.457*** | 0.461*** |
| | (27,273.46) | (352.70) | (346.05) | (363.72) | (319.78) | (332.04) | (280.52) | (344.73) | (60.85) | (50.48) |
| Observations | 27,839 | 27,839 | 27,839 | 27,839 | 27,839 | 27,839 | 27,839 | 27,839 | 27,839 | 27,839 |
| R-squared | 0.102 | 0.111 | 0.112 | 0.113 | 0.124 | 0.162 | 0.166 | 0.182 | 0.239 | 0.239 |
| Ind FE | YES | YES | YES | YES | YES | YES | YES | YES | YES | YES |
| Year FE | YES | YES | YES | YES | YES | YES | YES | YES | YES | YES |
| City FE | YES | YES | YES | YES | YES | YES | YES | YES | YES | YES |

Robust t-statistics in parentheses.

Note: *** p<0.01, ** p<0.05, * p<0.1.

1) Ind FE, Year FE, City FE denote fixed effect controls at the industry, year, and city level, respectively.

2) All in parentheses are t-statistics computed using robust standard errors clustered to the industry level.

corresponding actions. The results of the variable SOE indicate that state-controlled enterprises are an important part of the national economy, and are often asked to take on more social responsibility, and play the role of example and model in the industry.

## 4.2. Parallel trend test

An important condition for using DID is the assumption that the experimental and control groups have the same parallel trend beforehand [97]. In this study, it is important to ensure that different groups of firms show consistent trends in corporate ESG performance before implementing the EPCF. This study used the event study method [98–100] to identify parallel trends between the experimental and control groups ex-ante, and chose 2017 as the base period (the year before the intervention went into effect (citing articles that use the year before the intervention as the base period)), and the

results are shown in Fig 1. The results of the parallel trend test indicate that the positive event estimates for all periods in the ex-ante (2018 and earlier) period are included in the confidence interval at the 95% confidence level, and therefore the original hypothesis of a parallel trend in the ex ante period is not rejected at the 5% level for the experimental and control groups. More importantly, although issuing green bonds has a significant positive effect on ESG performance [53], the effect is not of such magnitude that there is an ex ante trend difference in ESG performance between the experimental and control groups. This is a side note that green bonds do not play a large role until they are widely recognized. More-over, in the ex-post (after 2018) event estimates, the estimates for 2019–2021 are all significantly positive at the 5% level. This implies that the results of the benchmark regressions are reliable and play a role in the second year of the interven-tion. This is because policy effects are not immediate but take time to fully materialize.

**4.3. Robustness test**

1) Replacement of explanatory variables

In order to mitigate the endogeneity problem caused by measurement error, we further verify the robustness of the conclu-sion that the EPCF can promote firms' ESG performance. Specifically, the ESG composite scores published by CNRDS (Chinese Research Data Services Platform) are used to replace the original explanatory variables for the stability test. The regression results of the replaced explanatory variables are shown in Table 3, **column (1)**. The estimation results show that the regression coefficients of the core explanatory variable, MIF, are still able to be significantly positive at the 5% level after replacing the proxies of the explanatory variables. This is able to further rule out the possibility of measurement error affecting the results.

2) Considering the lagged impact

The results considering the lagged effects are shown in Table 3, **column (2)**. The estimation results show that the regression coefficients of the core explanatory variables are significantly positive at the 5% level when we front-load the

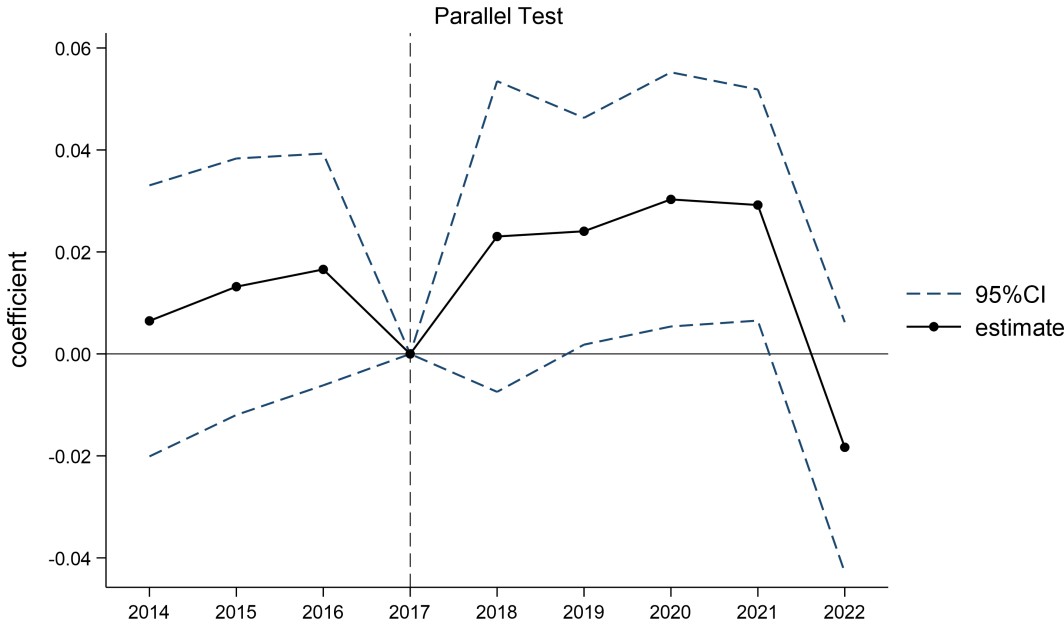

**Fig 1. Parallel test.**

**Table 3. Robustness test.**

| VARIABLES | (1) hzESG | (2) cnrdsESG | (3) F.ESG | (4) ESG | ESG |
|---|---|---|---|---|---|
| did | 0.020*** | 1.934** | 0.012** | 0.020** | 0.020*** |
| | (3.03) | (2.20) | (2.50) | (2.83) | (3.03) |
| DUA | -0.000 | -0.400** | -0.000 | -0.000 | -0.000 |
| | (-0.49) | (-2.71) | (-0.30) | (-0.47) | (-0.49) |
| SOE | 0.014*** | 0.209 | 0.013*** | 0.014*** | 0.014*** |
| | (6.53) | (1.00) | (6.55) | (6.22) | (6.53) |
| LNGDP | -0.000 | -0.043 | -0.000 | —— | -0.000 |
| | (-0.67) | (-0.55) | (-1.33) | | (-0.67) |
| LNAGE | -0.012*** | 0.597*** | -0.006*** | -0.012*** | -0.012*** |
| | (-16.56) | (6.80) | (-13.54) | (-16.26) | (-16.56) |
| LEV | -0.045*** | 1.536*** | -0.025*** | -0.045*** | -0.045*** |
| | (-17.43) | (4.55) | (-11.19) | (-18.49) | (-17.43) |
| GROWTH | -0.001 | 0.004 | 0.000 | -0.001 | -0.001 |
| | (-1.40) | (0.04) | (0.67) | (-1.27) | (-1.40) |
| SIZE | 0.014*** | 0.863*** | 0.010*** | 0.014*** | 0.014*** |
| | (37.64) | (11.23) | (17.83) | (34.96) | (37.64) |
| DOMINANCE | 0.000 | -0.006 | -0.000 | 0.000 | 0.000 |
| | (0.63) | (-1.27) | (-0.77) | (0.66) | (0.63) |
| ROA | 0.114*** | 0.421 | 0.133*** | 0.121*** | 0.114*** |
| | (17.36) | (0.27) | (13.68) | (19.33) | (17.36) |
| Constant | 0.461*** | 7.374*** | 0.533*** | 0.458*** | 0.461*** |
| | (50.48) | (3.28) | (42.52) | (49.27) | (50.48) |
| Observations | 27,839 | 27,839 | 22,766 | 27,832 | 27,839 |
| R-squared | 0.239 | 0.236 | 0.149 | 0.256 | 0.239 |
| Ind FE | YES | YES | YES | YES | YES |
| Year FE | YES | YES | YES | YES | YES |
| City FE | YES | YES | YES | YES | YES |
| Province FE | —— | —— | —— | YES | —— |
| Year$\times$Ind FE | —— | —— | —— | YES | —— |
| Year$\times$City FE | —— | —— | —— | YES | —— |

Note: *** $p<0.01$, ** $p<0.05$, * $p<0.1$, same below.

1)Province FE indicates fixed effect controls at the provincial level, the rest fixed effects are the same as in Table 2.

2)All in parentheses are t-statistics computed using robust standard errors clustered to the industry level.

explanatory variables by one period, but their estimates are significantly lower than the original 0.02. This result is also in line with the economic reality that the impact of the central bank's policy of expanding the collateral framework diminishes with the passage of time.

3) Consider more potential impact factors

Consider those external shocks that may have specific impacts on the agriculture and high-tech industries and some cities due to natural disasters (e.g., severe flooding, extraordinarily heavy rains in some cities in Guangdong, Fujian, and Henan provinces) and changes in the international trade environment (e.g., technological embargoes on the high-tech industry due to the U.S.-China trade war). This study fixes the industry-time, city-time interaction effects. The results after adding

the interaction fixed effects are shown in Table 3, **column (3)**. The regression coefficients of the core explanatory variables are significantly positive at the 5% level and their estimated coefficients are the same as in the main regression.

4) Perform Tailoring

In order to prevent outliers from influencing the study results, continuous variables are usually subjected to two-way shrinkage at the 1% and 99% quantiles. The specific results are shown in Table 3, **column (4)**, the regression coefficient of Policy is 0.020 and significant at 1% level, which is basically the same as that of the main regression.

## 4.4. Endogeneity test

### 4.4.1. Instrumental variables approach.
Since the formulation of the EPCF is a human decision and not entirely a random event, it is often influenced by factors such as existing markets. Therefore, the estimation results using the ordinary least squares method may have endogeneity problems.

In this paper, the mean debt issuance size of other firms within the same book (excluding itself) is selected as the instrumental variable in this paper to address the potential endogeneity problem. The correlation test and estimation results of the two-stage least squares method are shown in Table 4. The estimation results in the first stage indicate that the effect of the instrumental variables on the core explanatory variables is significant at the 1% level.

In the second-stage estimation results, the regression coefficient of the core explanatory variable *did* is significant at the 1 percent level at 0.0838. This means that, on average, firms that are affected by EPCF will improve their relative ESG performance by about 0.08 units. Thus, the results of the benchmark regression substantially underestimate the policy effect of the EPCF.

Furthermore, from Table 5, the results of the endogeneity test indicate a p-value of 0.0072, which is less than the 0.01 critical value. This indicates that the regression using ordinary least squares does have endogeneity problems. In addition, the non-identifiability test's indicate that it satisfies the rank condition (i.e., p-value is less than 0.001) and the F-statistic for the weak instrumental variable test is 56.054, which significantly rejects the original hypothesis that the instrumental variables are for weak instrumental variables.

1) The endogeneity test requires a p-value of less than 0.1 to be passed

2) The unidentifiable test uses the Kleibergen-Paap rk LM statistic with a value greater than 10 as a pass

3) The weak instrumental variable test used the Kleibergen-Paap rk Wald F statistic with a value greater than the 10% Stock-Yogo (16.38) threshold as a pass.

### 4.4.2. PSM-DID.
The DID model used in this study entails comparing the differences between the treatment group (the group affected by the policy) and the control group (the group not affected by the policy) before and after the implementation of the policy, but the characteristics of the treatment group and the control group are not exactly the same. Therefore, to avoid selection bias due to other factors. Cf. Heyman et al. (2007) [101], we match the experimental and control groups using nearest-neighbor one-to-many matching without put-back. Then, a test of between-group balance is performed and a propensity score (i.e., the probability that an individual will receive the treatment) is calculated for the control group to screen out individuals in the control group who are similar to the treatment group on several dimensions, thus reducing selection bias. The kernel density function curves before and after matching for the treatment and control groups are shown in Figs 2 and 3, respectively. As can be seen from the figures, after the matching was performed, the off-center trend and the mean value of the propensity score of the treatment group were significantly larger than those of the control group. And after matching, both the off-medium trend and the mean value of the propensity score of the treatment group are very close to those of the control group. This implies that without matching, a direct comparison of the differences between the two sample groups would lead to

**Table 4. Result of 2SLS.**

| VARIABLES | (1) | (2) |
|---|---|---|
| | First | Second |
| | Policy | hzESG |
| Meanscale | -10.96*** | |
| | (1.463) | |
| did | | 0.0838*** |
| | | (0.0179) |
| GROWTH | 0.000285 | -0.00102 |
| | (0.000443) | (0.000724) |
| DUA | 0.000672 | -0.000407 |
| | (0.00126) | (0.000719) |
| DOMINANCE | -3.85e-05 | 2.86e-05 |
| | (4.46e-05) | (4.08e-05) |
| ROA | -0.00201 | 0.114*** |
| | (0.00538) | (0.00656) |
| SIZE | 0.00344*** | 0.0136*** |
| | (0.00100) | (0.000346) |
| LNAGE | -0.000139 | -0.0124*** |
| | (0.000457) | (0.000739) |
| SOE | -0.00154 | 0.0143*** |
| | (0.00146) | (0.00216) |
| LNGDP | 0.000107 | -0.000307 |
| | (0.000182) | (0.000436) |
| LEV | 0.00492 | -0.0457*** |
| | (0.00356) | (0.00256) |
| Ind FE | YES | YES |
| Year FE | YES | YES |
| City FE | YES | YES |
| Observations | 27,839 | 27,839 |
| R-squared | | 0.148 |

1)The fixed effects are the same as in Table 2.

2)All in parentheses are t-statistics computed using robust standard errors clustered to the industry level.

serious errors. In addition, Fig 4 reports the results of the balance test. As can be seen in Fig 2, the gap between the groups for each covariate is significantly reduced after matching. Finally, the results of regressions using samples with non-null matching weights and satisfying the common support assumption are reported in columns (1) and (2) of Table 6, respectively, as well as the results of conducting frequency-weighted regressions in column (3). The results show that the regression coefficients for *did* are all significantly positive at the 5-percent level and the estimated coefficients are not significantly different from the baseline regression.

## 4.5. Placebo test

### 4.5.1. Randomized spatial placebo test.

To rule out that the above findings are subject to chance, k firms with the same number of experimental groups as in the previous section were randomly selected from the sample without put-back as pseudo-treated individuals and repeated 500 times for the baseline regression. The distribution of the estimated coefficients is shown in shown in Fig 5. As can be seen from Fig 3, the distribution of the placebo effect for the

**Table 5. Result of the instrumental variable test.**

| Endogeneity test of endogenous regressor | | p=0.0072 |
|---|---|---|
| Underidentification test [p value] | 4.61[0.0318] | |
| Weak identification test | | F=56.054 |
| Ind FE | YES | YES |
| Year FE | YES | YES |
| City FE | YES | YES |
| Observations | 27,839 | 27,839 |
| R-squared | | 0.148 |

1) The endogeneity test requires a p-value of less than 0.1 to be passed.

2) The unidentifiable test uses the Kleibergen-Paap rk LM statistic with a value greater than 10 as a pass.

3) The weak instrumental variable test used the Kleibergen-Paap rk Wald F statistic with a value greater than the 10% Stock-Yogo (16.38) threshold as a pass.

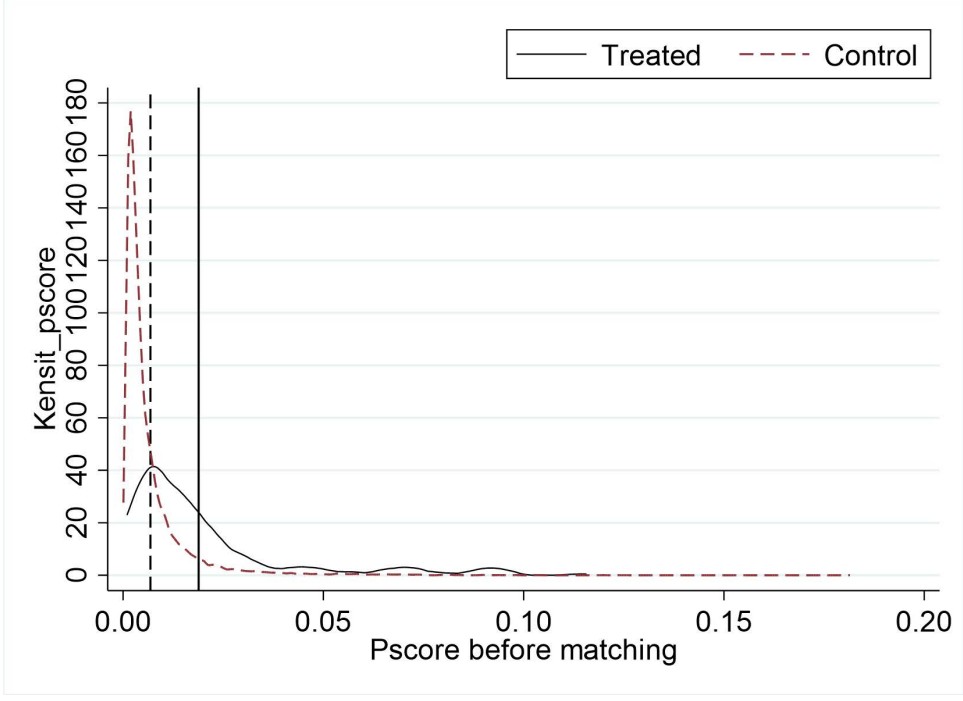

**Fig 2. Pscore before matching.**

randomized sample roughly exhibits a normal distribution with 0 as the expectation. In contrast, the estimated coefficient for the benchmark regression is approximately 0.02, which deviates substantially from the center of the distribution posed by the random sample. Thus, the probability that the outcome of the benchmark regression occurred by chance is very small.

**4.5.2. Mixed placebo test.** To avoid chance that the results of the baseline regression are also confounded by time-level randomization, further consideration was given to randomly selecting the intervention period during the random sampling process and repeating the same 500 estimates. The distribution of the estimates for the mixed placebo is shown

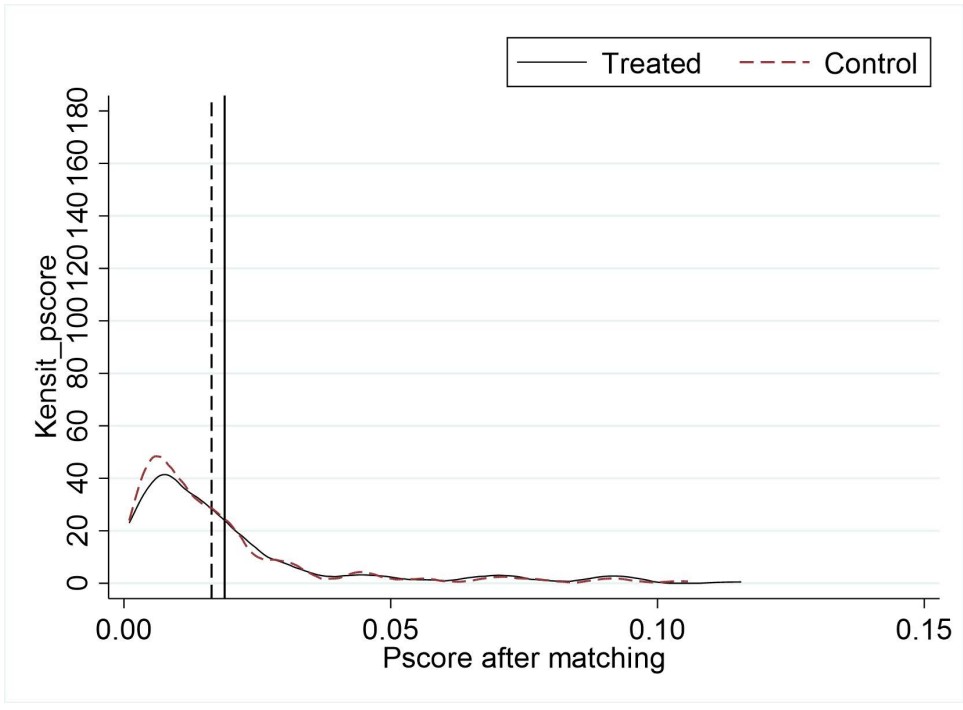

**Fig 3. Pscore after matching.**

in Fig 6. The actual estimates lie in the right-hand tail of the distribution of the placebo effect. This further suggests that the probability that the outcome of the benchmark regression is due to chance at any level is extremely small. Thus, the conclusions of Hypothesis 1 are again verified as robust.

### 4.6. Heterogeneity analysis

1) Heterogeneity in the degree of pollution caused by firms

The main purpose of the central bank's inclusion of green bonds in the collateralized collateral framework is to alleviate the financial constraints for enterprises to engage in green production in order to encourage them to engage in sustainable development. However, enterprises that cause different degrees of environmental pollution will face different costs and difficulties in practicing the ESG concept. For enterprises in heavily polluting industries such as petroleum processing, metal smelting, and chemical manufacturing, the process of carrying out green transformation often requires a large amount of capital and time due to the characteristics of the industry and is accompanied by great risks. As a result, heavily polluting firms do not share the goals of most profit-avoiding investors, which leads to the fact that these firms may raise far less green capital than non-heavily polluting firms. Finally, the gap in the funds used for green transformation will lead to a less effective enhancement of the ESG performance of heavily polluting firms than that of non-heavily polluting firms. The results of the firm heterogeneity analysis based on pollution degree are shown in Table 7, **column (1)**. The results show that the regression coefficient of the cross-multiplier term of the dummy variable polluted and ***did***, which distinguishes the degree of heavy pollution of firms, is significantly negative at the 5% level, which verifies the previous speculation.

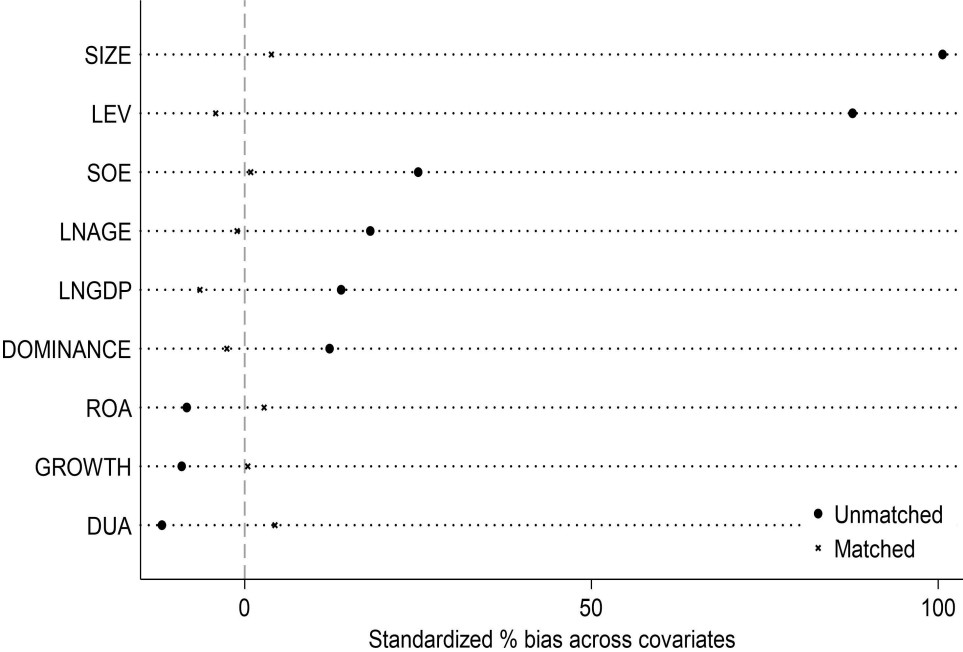

**Fig 4. Balance test.**

2) Heterogeneity of marketization level

In regions with a high level of marketization, their laws and regulations for the environment tend to be more stringent, and firms need to pay more attention to their ESG practices in order to ensure compliance. Finally, the difference in the degree of strictness of control will lead to the fact that the ESG performance of enterprises located in regions with a low level of marketization will not be as effective as that of enterprises located in regions with a high level of marketization. The results of the analysis of firm heterogeneity based on the level of marketization are presented in Table 7, **column (2)**. The results show that the regression coefficients of the cross-multiplier terms of the dummy variables market and *did*, which distinguish firms' marketization levels, are significantly positive at the 5% level, which verifies the previous speculation.

3) Heterogeneity of geographic location

Coastal regions have richer transportation modes and are more connected to global supply chains, which have higher standards for ESG in order to maintain relationships with partners and ensure supply chain sustainability, firms in coastal regions pay more attention to and improve their ESG performance. Finally, due to the different ESG standards of their partners, non-coastal firms do not improve their ESG performance as much as coastal firms. The results of the firm heterogeneity analysis based on the heterogeneity of geographic location are presented in column (3) of Table 7, **column (3)**. The results show that the regression coefficients of the cross-multiplication terms of the dummy variables location and *did*, which distinguish geographic location, are significantly positive at the 5% level, which validates the previous conjectures.

### 4.7. Impact mechanism test

**4.7.1. Financing constraints.** The implementation of the EPCF has significantly broadened enterprises' financing channels, enabling more types of assets to be used as eligible collateral. This not only makes it easier for firms to obtain

**Table 6. Regression after matching.**

| VARIABLES | (1) | (2) | (3) |
|---|---|---|---|
| | hzESG | hzESG | hzESG |
| did | 0.019** | 0.019*** | 0.017** |
| | (2.54) | (2.97) | (2.53) |
| LNAGE | -0.006** | -0.012*** | -0.006** |
| | (-2.90) | (-15.81) | (-2.90) |
| LEV | -0.053*** | -0.047*** | -0.051*** |
| | (-3.50) | (-15.53) | (-3.43) |
| GROWTH | 0.001 | -0.001 | 0.001 |
| | (0.41) | (-1.36) | (0.45) |
| SIZE | 0.011*** | 0.014*** | 0.012*** |
| | (8.65) | (36.54) | (9.90) |
| DUA | 0.003 | -0.000 | 0.002 |
| | (0.78) | (-0.68) | (0.60) |
| DOMINANCE | -0.000 | 0.000 | -0.000 |
| | (-0.20) | (0.44) | (-0.33) |
| ROA | 0.187*** | 0.115*** | 0.198*** |
| | (3.96) | (17.66) | (4.26) |
| SOE | 0.016*** | 0.014*** | 0.015*** |
| | (5.95) | (7.08) | (5.68) |
| LNGDP | -0.001 | -0.000 | -0.001 |
| | (-1.02) | (-0.87) | (-0.80) |
| Constant | 0.531*** | 0.465*** | 0.503*** |
| | (14.42) | (49.15) | (14.12) |
| Observations | 1,034 | 26,197 | 1,092 |
| R-squared | 0.370 | 0.239 | 0.377 |
| Ind FE | YES | YES | YES |
| Year FE | YES | YES | YES |
| City FE | YES | YES | YES |

1) The rest fixed effects are the same as in Table 2.

2) All in parentheses are t-statistics computed using robust standard errors clustered to the industry level.

bank loans and other forms of financing support, but also, due to the diversification of collateral, banks have more references when assessing the risk of lending. This change has helped to reduce banks' lending premiums for high-risk enterprises, which in turn has effectively lowered the cost of financing for enterprises. In addition, by flexibly adjusting the collateral policy, the central bank is able to actively guide commercial banks to increase credit investment in specific areas (such as green industry, social responsibility programs, etc.), thus effectively alleviating the financing constraints in these key areas. On this basis, this paper constructs a mechanism analysis, as shown in (1), and its results are shown in Table 8. The regression coefficients of the core explanatory variables *did* are as follows. The regression coefficient of the core explanatory variable *did* is significantly positive at the 1% level, which indicates that the EPCF is able to alleviate the financing constraints of firms, which in turn enhances their ESG performance. This verifies hypothesis h2a.

**4.7.2. Level of green innovation.** In the previous heterogeneity analysis, we have demonstrated that there is a difference in the degree to which the ESG performance of non-heavily polluting firms is affected by the EPCF relative to that of heavily polluting firms. Green innovation of enterprises related to corporate pollution also plays a certain role. On this basis, enterprises supported by the EPCF are more likely to be recognized by investors and consumers in the

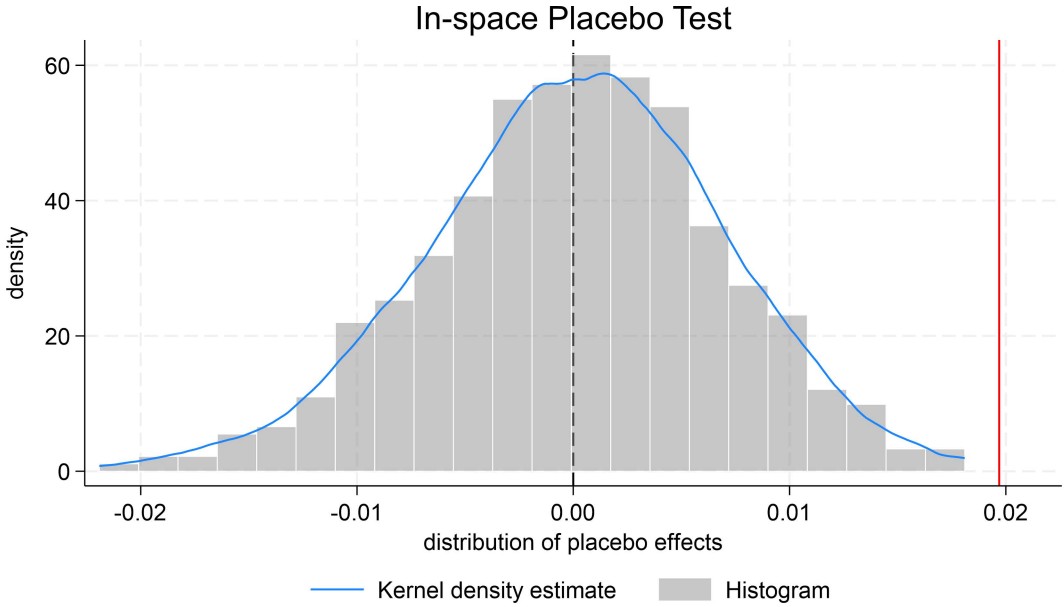

**Fig 5. In-space placebo test.**

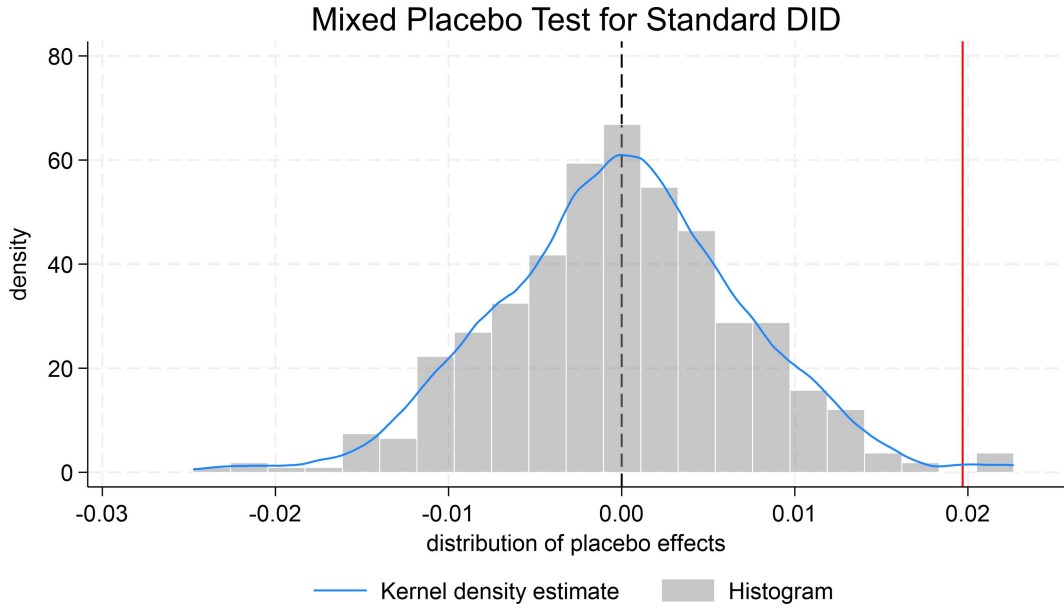

**Fig 6. Mixed placebo test for standard DID.**

market, and this recognition enhancement can motivate enterprises to increase their investment in green innovation to meet the market's demand for environmental protection and sustainability, which in turn enhances the ESG performance of enterprises. On this basis, this paper constructs a mechanism analysis, and in the process of replacing the explanatory variables with mechanism variables, its value is added by 1 at the same time and made logarithmic treatment. As shown

**Table 7. Heterogeneity analysis.**

| VARIABLES | (1) | (2) | (3) | (4) |
|---|---|---|---|---|
| | hzESG | hzESG | hzESG | hzESG |
| did×polluted | —— | -0.026** | —— | —— |
| | | (-2.62) | | |
| did×market | —— | —— | 0.019** | —— |
| | | | (2.61) | |
| did×location | —— | —— | —— | 0.027** |
| | | | | (2.37) |
| did | 0.019*** | —— | —— | —— |
| | (2.99) | | | |
| polluted | 0.006*** | —— | —— | —— |
| | (3.25) | | | |
| market | -0.002** | —— | —— | —— |
| | (-2.34) | | | |
| location | 0.002 | —— | —— | —— |
| | (1.64) | | | |
| Constant | 0.460*** | 0.461*** | 0.461*** | 0.461*** |
| | (48.22) | (50.24) | (50.16) | (49.06) |
| Covariate | Control | Control | Control | Control |
| Observations | 27,834 | 27,839 | 27,839 | 27,834 |
| R-squared | 0.240 | 0.240 | 0.239 | 0.239 |
| Ind FE | YES | YES | YES | YES |
| Year FE | YES | YES | YES | YES |
| City FE | YES | YES | YES | YES |

1)The rest fixed effects are the same as in Table 2.

2)All in parentheses are t-statistics computed using robust standard errors clustered to the industry level.

**Table 8. Mechanism testing.**

| VARIABLES | (1) | (2) |
|---|---|---|
| | GREENTECH | SA |
| did | 0.719*** | 0.079*** |
| | (3.93) | (3.54) |
| Constant | -4.177*** | -4.728*** |
| | (-3.04) | (-20.31) |
| Control vaviables | YES | YES |
| Observations | 27,839 | 27,839 |
| R-squared | 0.218 | 0.382 |
| Ind FE | YES | YES |
| Year FE | YES | YES |
| City FE | YES | YES |

1) The rest fixed effects are the same as in Table 2.

2) All in parentheses are t-statistics computed using robust standard errors clustered to the industry level.

in (2), and its results are shown in Table 8. The regression coefficient of the core explanatory variable **did** is significantly positive at the 1% level, which indicates that the EPCF promotes the level of green innovation of firms, which in turn enhances their ESG performance. This verifies hypothesis h2b.

## 5. Conclusion and recommendations

The central bank serves as a fundamental authority in the regulation and control of macroeconomic policies, and the implementation of its associated policies is crucial for the formulation of corporate sustainable development strategies. This study aims to investigate whether the expansion of the central bank's collateral framework for green bond policies can facilitate corporate sustainable development. To this end, corporate ESG (Environmental, Social, and Governance) performance is utilized as the evaluative metric for corporate sustainable development. The analysis focuses on A-share listed companies in mainland China from 2014 to 2022, employing methodologies such as Difference-in-Differences (DID) and Propensity Score Matching Difference-in-Differences (PSM-DID). The findings of this research are as follows: First, the EPCF significantly enhances corporate ESG performance and successfully passes the parallel trend test. Robustness checks, which include substituting explanatory variables, accounting for the lagged effects, considering additional potential influencing factors, and trimming the tails, confirm the sustained impact of EPCF on firms' ESG performance. Second, the heterogeneity analysis reveals that the EPCF exerts a more substantial influence on the ESG performance of non-heavily polluting firms compared to heavily polluting firms. Furthermore, the EPCF is found to enhance ESG performance more significantly in regions characterized by high levels of marketization than in those with low marketization levels, indicating that the transmission of the policy is more effective in market-oriented regions. Additionally, the policy demonstrates a more pronounced positive effect on corporate ESG performance in coastal regions compared to non-coastal regions. This suggests that the impact of the EPCF on corporate ESG performance is influenced by various factors, necessitating tailored policy approaches. Lastly, the mechanism analysis indicates that EPCF can enhance corporate ESG performance by alleviating financing constraints and fostering corporate green innovation. The policy improves corporate access to credit, reduces financing costs, and optimizes capital allocation, thereby promoting advancements in green technology research and development within enterprises.

## 6. Research limitations and outlook

The EPCF is effective in enhancing corporate environmental, social, and governance (ESG) performance. The primary mechanism through which this policy operates is by indirectly increasing the value of eligible collateral through the enhancement of its scarcity. When access to qualified collateral is limited, an excess of such collateral can lead to a dilution of its value, resulting in a corresponding reduction in the central bank's willingness to lend. This scenario can weaken the overall effectiveness of the policy. Furthermore, the impact of the EPCF on firms' ESG performance may also be diminished. Consequently, it is imperative to improve the dynamic monitoring and assessment system of collateral to ensure effective risk management.

In the ongoing effort to optimize the collateral framework of the central bank, it is essential to recognize the substantial disparity in the financial requirements for the green transformation of heavy polluters compared to non-heavy polluters. To address this, it is imperative to leverage the comparatively lower costs associated with the green transformation of non-heavy polluters while simultaneously providing subsidies to facilitate the green transformation of heavy polluters. Furthermore, enterprises operating in regions characterized by high levels of marketization should be encouraged to enhance their Environmental, Social, and Governance (ESG) information disclosure. This can be achieved by transparently demonstrating their ESG performance and improvement measures through open and transparent information. In regions with a low level of marketization, the publicity of the collateral framework expansion policy should be strengthened through multiple channels to increase enterprises' awareness of the policy. Coastal enterprises are more closely connected to global supply chains and

 

have stricter ESG standards, so coastal enterprises should continue to improve their ESG standards to meet the needs of overseas customers and enhance their corporate image, while non-coastal enterprises should learn from the success stories and experiences of coastal enterprises in ESG and continue to improve their understanding of ESG. Finally, the central bank should take concrete measures to encourage financial institutions to lend to eligible enterprises, such as releasing more liquidity by lowering the reserve requirement ratio. In addition, banks can be incentivized to increase the amount of green loans to eligible enterprises by providing them with a risk protection mechanism to reduce their risk of runs. The government should introduce a series of policy measures to support enterprises in green innovation. First, it can produce and release promotional videos to raise awareness of the importance of green innovation throughout society. Second, the government can formulate new policies, such as setting up a sustainable development recognition mechanism and granting preferential policies, such as tax breaks and capital subsidies, to enterprises that meet the recognition criteria, in order to reduce the cost of green innovation for enterprises. More importantly, the government can also provide these enterprises with official promotion, credit guarantee and project cooperation support to enhance their credibility and competitiveness in the market.

It is essential to acknowledge that, while the current analysis offers significant insights, there are limitations to its applicability. This study specifically examines the expansion of green bond collateral in China; however, variations in geography, economic conditions, policy frameworks, and market dynamics across different countries and regions may influence the generalizability of the findings. Consequently, this limitation precludes the direct extrapolation of the conclusions to other countries or regions globally. Furthermore, our concentrated focus on green bonds as a distinct category of collateral—an approach that facilitates an understanding of their eventual impacts on the sustainability sector—results in a lack of direct evidence regarding the potential effects of other collateral types. This research gap indicates that the validity and applicability of various collateral types and study samples warrant further exploration and verification to comprehensively understand the broader implications of the EPCF on the financial market as a whole.

## Supporting information

**S1 Data. Data of manuscript.**
(XLSX)

## Author contributions

**Conceptualization:** Xupei Wang.

**Data curation:** Xupei Wang.

**Formal analysis:** Xupei Wang.

**Funding acquisition:** Liang Zhao.

**Investigation:** Xupei Wang.

**Methodology:** Xupei Wang.

**Project administration:** Xupei Wang.

**Resources:** Xupei Wang.

**Software:** Xupei Wang.

**Supervision:** Xupei Wang, Liang Zhao.

**Validation:** Xupei Wang.

**Visualization:** Xupei Wang.

**Writing – original draft:** Xupei Wang.

**Writing – review & editing:** Xupei Wang.

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
