## [Decision Letter · Decision Letter 0]

7 Jan 2025

PONE-D-24-49919ESG Growth Catalyst: China's Central Bank Collateral framework expansionPLOS ONE

Dear Dr. Wang,

Thank you for submitting your manuscript to PLOS ONE. After careful consideration, we feel that it has merit but does not fully meet PLOS ONE’s publication criteria as it currently stands. Therefore, we invite you to submit a revised version of the manuscript that addresses the points raised during the review process.

We look forward to receiving your revised manuscript.

Kind regards,

Wafa Ghardallou

Academic Editor

PLOS ONE

2. Thank you for stating the following financial disclosure:  [Guangdong Province Philosophy and Social Science Planning Project (GD24XWL01)�Research project funded by the Guangdong Ocean University Scientific Research Startup Fund: "Study on the Transmission Mechanism of Imported Financial Crises"]. 

5. Please include a caption for figure 1, 2A, 3, 4 and 5.

Reviewers' comments:

Reviewer's Responses to Questions

**Comments to the Author**

1. Is the manuscript technically sound, and do the data support the conclusions?

Reviewer #1: Partly

Reviewer #2: Yes

2. Has the statistical analysis been performed appropriately and rigorously? 

Reviewer #1: Yes

Reviewer #2: Yes

3. Have the authors made all data underlying the findings in their manuscript fully available?

Reviewer #1: Yes

Reviewer #2: Yes

4. Is the manuscript presented in an intelligible fashion and written in standard English?

Reviewer #1: No

Reviewer #2: Yes

5. Review Comments to the Author

Reviewer #1: The paper's topic and conducted research are very important and justified to be presented in a high-quality Journal. The subject is very important for the literature. The paper is informative, but some issues need to be addressed carefully. My decision is – a major revision, with some amendments. Please see my comments and suggestions below.

Comment 1. The Introduction needs to be rearranged, and more work is needed to strengthen the theoretical basis. Additionally, how this study bridges the research gaps should be elucidated. The following paper can be a good example to help you improve your paper (Does firm‐level exposure to climate change influence inward foreign direct investment? Revealing the moderating role of ESG performance. Corporate Social Responsibility and Environmental Management. https://doi.org/10.1002/csr.2917).

Comment 2. More existing literature should be presented in the introduction. The following paper can be a good example to help you improve your paper (Effect of ESG Performance on Corporate Green Technology Innovation: The Mediating Role of Financial Constraints and Digital Transformation. The Singapore Economic Review, 1-26.).

Comment 3. Also, please add the structure of this study at the end of the introduction.

Comment 4. The manuscript falls short in terms of theoretical development. What theories can be used as the theoretical basis for your research topic?

Comment 5. The arguments proposed for Hypotheses 1, 2a and 2b are insufficient to support the proposition of these hypotheses. The authors should address the arguments for each hypothesis separately. Also, Hypothesis 3c should be supported by more existing literature.

Comment 6. The theoretical model should be presented in section 2.

Comment 7. The authors should add the section conclusion.

Comment 8. The language of this paper is very bad and needs help from native speakers.

Good luck for your work!

Reviewer #2: The author(s) has/have done well in addressing an important research topic entitled ‘ESG Growth Catalyst: China's Central Bank Collateral framework expansion. However I have the following observations and suggestion that can improve the quality of the paper.

1. The abstract should clearly highlight the major findings of the study

2. Explanation about the contribution of the paper needs to be improved.

3. While developing the hypotheses, the author(s) did not provide proper arguments and synthesize the literature. Generally, the literature review has not been locally and extensively discussed with many unexpected hypotheses

4. There is a need for the authors to clearly explain the reasons for including green innovation of corporations in the paper.

5. The authors need to discuss and justify the period and the sample they used in the methodology

6. What is the meaning of 5 Discussion in the paper? Is it written in the right place? Please, check

7. There are many typographical errors in the manuscript for instance; (in the abstract ‘be-tween’) (first paragraph of introduction (chain ++management). There is a need for a serious proofreading of the paper

6. PLOS authors have the option to publish the peer review history of their article (what does this mean? ). If published, this will include your full peer review and any attached files.

**Do you want your identity to be public for this peer review?** For information about this choice, including consent withdrawal, please see our Privacy Policy .

Reviewer #1: No

Reviewer #2: No

---

## [Author Response · Author response to Decision Letter 1]

13 Mar 2025

We feel great thanks for your professional review work on our article. As you are concerned, there are several problems that need to be addressed. According to your nice suggestion, we have made extensive corrections to our previous manuscript, the detailed corrections are listed below.

Reply to Reviewer 1's suggestion:

Comment 1. The Introduction needs to be rearranged, and more work is needed to strengthen the theoretical basis. Additionally, how this study bridges the research gaps should be elucidated. The following paper can be a good example to help you improve your paper (Does firm‐level exposure to climate change influence inward foreign direct investment? Revealing the moderating role of ESG performance. Corporate Social Responsibility and Environmental Management. https://doi.org/10.1002/csr.2917).

Our response: First of all, thank you very much for reviewing our work. We think this is a good suggestion. In the reference (Does firm‐level exposure to climate change influence inward foreign direct investment? Revealing the moderating role of ESG performance. Corporate Social Responsibility and After Environmental Management. https://doi.org/10.1002/csr.2917) this paper, we use more literature reorganize the introduction. In addition, we supplement the limitations of the existing research in the introduction, and thus illustrate how our study fills the research gap.

Comment 2. More existing literature should be presented in the introduction. The following paper can be a good example to help you improve your paper (Effect of ESG Performance on Corporate Green Technology Innovation: The Mediating Role of Financial Constraints and Digital Transformation. The Singapore Economic Review, 1-26.).

Our response: We sincerely thank you for your valuable comments. In reference to what you mentioned (Effect of ESG Performance on Corporate Green Technology Innovation: The Mediating Role of Financial Constraints and Digital Transformation. The Singapore Economic Review, 1-26.) After this paper, We have added more literature to the introduction of the revision as evidence. In addition, we cite this paper as an argument in the process of demonstrating the role of enterprises in green technology innovation as a mechanism.

Comment 3. Also, please add the structure of this study at the end of the introduction.

Our response: Thank you for your reminder. We have supplemented the description of the various parts of this study in the last paragraph of the introduction.

Comment 4. The manuscript falls short in terms of theoretical development. What theories can be used as the theoretical basis for your research topic?

Our response: This is a very valuable suggestion and we take it very seriously. After repeatedly consulting relevant literature and books, we supplemented the scarcity theory and information asymmetry theory in Part 2.2 to put forward the hypothesis of our research.

Comment 5. The arguments proposed for Hypotheses 1, 2a and 2b are insufficient to support the proposition of these hypotheses. The authors should address the arguments for each hypothesis separately. Also, Hypothesis 3c should be supported by more existing literature.

Our response: Thank you for pointing this out. Hypothesis 1, 2a, and 2b are indeed inadequate. Therefore, we add more literature as evidence in hypothesis 1, 2a and 2b respectively, and pay more attention to the logical relationship in the discussion. In addition, as to the suggestion that hypothesis 3c needs more arguments, we propose hypothesis 3c based on the perspective of industry spillover and external supervision, and add more literature as arguments.

Comment 6. The theoretical model should be presented in section 2.

Our response: This is a very good suggestion, but because we are very weak in this area, we can not build a suitable mathematical model to carry out theoretical analysis.

Comment 7. The authors should add the section conclusion.

Our response: Thanks for your suggestion, we have added a conclusion section to Section 5 of our study.

Comment 8. The language of this paper is very bad and needs help from native speakers.

Our response: Thanks for your suggestion We feel sorry for our poor writings, however, we do invite a friend of us who is a native English speaker to help polish our article. And we hope the revised manuscript could be acceptable for you.

Reply to Reviewer 2's suggestion:

1. The abstract should clearly highlight the major findings of the study

Our response: First of all, thank you very much for reviewing our work. This is very good advice. Therefore, we highlight the keyword "research results show" in the abstract, and the impact of EPCF on enterprise ESG performance is expressed numerically, so that readers can understand our research more intuitively.

2. Explanation about the contribution of the paper needs to be improved.

Our response: Thank you very much for your suggestion. We have rewritten the marginal contribution of our study in the revised citation section

3. While developing the hypotheses, the author(s) did not provide proper arguments and synthesize the literature. Generally, the literature review has not been locally and extensively discussed with many unexpected hypotheses

Our response: Thank you for your advice. We have added a lot of literature as evidence and restated our hypothesis based on scarcity theory and information asymmetry theory.

4. There is a need for the authors to clearly explain the reasons for including green innovation of corporations in the paper.

Our response: This proposal is well worth our adoption. The inclusion of corporate green innovation in the study not only helps to reveal the mechanism of EPCF policy's impact on corporate ESG performance, but also provides more targeted suggestions for policy makers. We have provided relevant arguments and arguments for considering corporate green innovation in the revised version of the literature Review section 2.2 Financing constraints and the mechanism role of green innovation, and described the process of its mechanism role.

5. The authors need to discuss and justify the period and the sample they used in the methodology

Our response: This suggestion helps to improve the rigor of our research. We have supplemented the discussion on the study period and sample rationality in section 3.3sample selection and data sources.

6. What is the meaning of 5 Discussion in the paper? Is it written in the right place? Please, check

Our response: Thank you for your valuable advice. We have split chapter 5 into two chapters. They are 5.conclusion and recommendations and 6.Research limitations and outlook.

7. There are many typographical errors in the manuscript for instance; (in the abstract ‘be-tween’) (first paragraph of introduction (chain ++management). There is a need for a serious proofreading of the paper

Our response: Thank you very much for pointing this out. We apologize for these oversights and promise not to repeat them in future submissions. Thank you again for your careful inspection. Sincerely,

Xupei Wang

Xupeiwang1021@126.com

---

## [Editor Report · Decision Letter 1]

15 Apr 2025

ESG Growth Catalyst: China's Central Bank Collateral framework expansion

PONE-D-24-49919R1

Dear Dr. Zhao,

We’re pleased to inform you that your manuscript has been judged scientifically suitable for publication and will be formally accepted for publication once it meets all outstanding technical requirements.

Kind regards,

Wafa Ghardallou

Academic Editor

PLOS ONE
---

## [Editor Report · Acceptance letter]

PONE-D-24-49919R1

PLOS ONE

Dear Dr. Zhao,

I'm pleased to inform you that your manuscript has been deemed suitable for publication in PLOS ONE. Congratulations! Your manuscript is now being handed over to our production team.

Kind regards,

on behalf of

Dr. Wafa Ghardallou

Academic Editor

PLOS ONE